# The Serum Uric Acid Level Is Related to the More Severe Renal Histopathology of Female IgA Nephropathy Patients

**DOI:** 10.3390/jcm10091885

**Published:** 2021-04-27

**Authors:** Won Jung Choi, Yu A Hong, Ji Won Min, Eun Sil Koh, Hyung Duk Kim, Tae Hyun Ban, Young Soo Kim, Yong Kyun Kim, Seok Joon Shin, Seok Young Kim, Young Ok Kim, Chul Woo Yang, Yoon-Kyung Chang

**Affiliations:** 1Department of Internal Medicine, Daejeon St. Mary’s Hospital, Catholic University of Korea, Daejeon 34943, Korea; 1jungchoi@gmail.com (W.J.C.); amorfati@catholic.ac.kr (Y.A.H.); alterego54@catholic.ac.kr (S.Y.K.); 2Department of Internal Medicine, Bucheon St. Mary’s Hospital, College of Medicine, The Catholic University of Korea, Bucheon 14647, Korea; blueberi12@gmail.com; 3Department of Internal Medicine, Yeouido St. Mary’s Hospital, College of Medicine, The Catholic University of Korea, Seoul 07345, Korea; fiji79@catholic.ac.kr; 4Department of Internal Medicine, Seoul St. Mary’s Hospital, College of Medicine, The Catholic University of Korea, Seoul 06591, Korea; scamph@catholic.ac.kr (H.D.K.); yangch@catholic.ac.kr (C.W.Y.); 5Department of Internal Medicine, Eunpyeong St. Mary’s Hospital, College of Medicine, The Catholic University of Korea, Eunpyeong 03476, Korea; deux0123@catholic.ac.kr; 6Department of Internal Medicine, Uijeongbu St. Mary’s Hospital, College of Medicine, The Catholic University of Korea, Uijeongbu 11765, Korea; dr52916@catholic.ac.kr (Y.S.K.); cmckyo@catholic.ac.kr (Y.O.K.); 7Department of Internal Medicine, St. Vincent’s Hospital, College of Medicine, The Catholic University of Korea, Suwon 16247, Korea; drkimyk@catholic.ac.kr; 8Department of Internal Medicine, Incheon St. Mary’s Hospital, College of Medicine, The Catholic University of Korea, Incheon 22711, Korea; imkidney@catholic.ac.kr

**Keywords:** IgA nephropathy, uric acid, glomerular sclerosis, mesangial matrix expansion, kidney biopsy

## Abstract

Hyperuricemia is a significant risk factor for cardiovascular morbidity and chronic kidney disease progression. IgA nephropathy (IgAN) is a well-known primary glomerular nephropathy. Hyperuricemia is associated with a poor prognosis in IgAN patients. We evaluated the association of hyperuricemia with the histopathological severity of IgAN in male and female patients; 658 patients diagnosed with IgAN via kidney biopsy were initially included. Baseline patient data were collected by eight university hospitals affiliated with the College of Medicine of the Catholic University of Korea. Pathological features were independently evaluated by eight expert pathologists working in the hospitals, and the consensus was reached. Of the initial 658 patients, 517 were finally included (253 males and 264 females). Hyperuricemia was defined as a serum uric acid (UA) level >7.0 mg/dL for males and >5.6 mg/dL for females; 108 (42.7%) males and 95 (35.9%) females exhibited hyperuricemia. Compared to the patients with normal UA levels, the global glomerulosclerosis, segmental sclerosis, mesangial matrix expansion (MME), endocapillary proliferation (ECP), interstitial fibrosis (IF), and tubular atrophy (TA) scores were higher in hyperuricemic males and females. In multivariable linear regression, the serum UA level correlated significantly with the MME, ECP, IF, and TA scores of female IgAN patients only.

## 1. Introduction

Immunoglobulin A nephropathy (IgAN) is the most common primary glomerular nephropathy, both in Korea and worldwide [1]. The clinical outcomes include asymptomatic hematuria, proteinuria, kidney failure, and even end-stage renal disease (ESRD) requiring renal replacement therapy [2,3,4]. Approximately 20–40% of IgAN nephropathy patients progress to ESRD within 10–20 years of diagnosis [5,6]. The prevalence of IgAN differs between males and females. The male:female ratio is almost 1:1 in Asian populations, compared to 6:1 in Europe and the United States [7,8,9]. The risk factors for IgAN progression to chronic renal failure or ESRD include an elevated serum creatinine level, hypertension, proteinuria, dyslipidemia, and hyperuricemia [7,10].

The definition of hyperuricemia differs between males and females. Females usually have lower serum uric acid (UA) levels than men. Male sex is a significant risk factor for hyperuricemia and gout; males have been affected four-fold more frequently than females [11,12,13,14]. Hyperuricemia predicts mortality from heart failure, cerebrovascular and cardiac ischemic events, hypertension, and metabolic syndrome. An elevated serum UA level has been shown to be an independent risk factor for chronic kidney disease and renal function failure [15,16,17]. In IgAN patients, hyperuricemia was an independent risk factor for a poor renal outcomes and all-cause mortality [18]. Both histopathological parameters and the serum UA level play roles in IgAN; hyperuricemia is associated with tubular atrophy (TA) and interstitial fibrosis (IF) [19,20]. Nagasawa et al. and Oh et al. recently reported that hyperuricemic females with IgAN had a worse prognosis than males [21,22]. We aimed to determine whether hyperuricemia was associated with more severe IgAN histopathological features in females than males.

## 2. Materials and Methods

### 2.1. Study Design and Data

This multicenter cross-sectional cohort study recruited IgAN patients who underwent kidney biopsies between January 2015 and May 2020 in eight university hospitals affiliated with the College of Medicine of the Catholic University of Korea. The study adhered to all relevant tenets of the Declaration of Helsinki and was approved by the Institutional Review Board of the College of Medicine, Catholic University of Korea (IRB no. XC19OEDI0025). Written informed consent was obtained from all patients before the biopsy. A total of 658 IgAN patients diagnosed via kidney biopsy were included. We excluded patients aged < 18 years, as well as 6 patients whose serum UA levels were not measured, and 134 patients whose histological data were incomplete or who had insufficient glomeruli for a diagnosis to be made.

### 2.2. Data Collection, Definitions, and Measurements

Kidney biopsy data were uploaded to the Kidney Biopsy Registry of the Catholic Medical Center, as were baseline clinical and laboratory data. Hyperuricemia was defined as a serum UA level > 7.0 mg/dL for males and >5.6 mg/dL for females [23]. Proteinuria was assessed via a spot test that yielded the urine protein:creatinine ratio [24]. The estimated glomerular filtration rate (eGFR) was calculated using the equation from the modification of diet in renal disease study [25]. Urinary red blood cell (RBC) levels were graded as follows: <3 RBCs/high-power field (HPF), 0; 3–5 RBCs/HPF, 1; 5–9 RBCs/HPF, 2; 10–19 RBCs/HPF, 3; and >19 RBCs/HPF, 4.

### 2.3. Histopathological Parameters

Renal samples were reviewed by eight expert pathologists working in the eight hospitals using the following grading system. Under a light microscope, global glomerulosclerosis (GS), segmental sclerosis (SS), and crescent formation (or capsular adhesion, CA) were calculated as percentages (involved glomeruli/total glomeruli). Mesangial matrix expansion (MME), mesangial cell proliferation (MCP), endocapillary proliferation (ECP), interstitial fibrosis (IF), and tubular atrophy (TA) were all graded from 0 to 4, as follows: 0, absent; 1, trace (<20%); 2, mild (20–40%); 3, moderate (40–70%); and 4, severe (≥70%). Based on immunofluorescence microscopy, the extent of mesangial deposition of IgA, C3, and C4d was graded as 0 (absent), +1 (trace), +2 (mild), +3 (moderate), or +4 (marked). Renal biopsy data were also assessed using the World Health Organization classification (grades I–VI) for IgA nephropathy [26].

### 2.4. Statistical Analysis

Continuous variables were compared using the *t*-test. Categorical variables are expressed as numbers with percentages and were compared using the chi-squared test. Linear regression analysis was performed to evaluate the associations between the serum UA level and histopathological parameters stratified using clinical and laboratory scoring systems. A *p*-value < 0.05 was considered statistically significant. All statistical analyses were performed using SPSS software (ver. 23.0; IBM Corporation, New York, NY, USA).

## 3. Results

### 3.1. Baseline Characteristics

A total of 517 patients were retrospectively analyzed (Figure 1). Patients with normal UA levels were compared to hyperuricemic patients. The mean patient age was 42.0 ± 14.6 years. There were 253 (48.9%) males and 264 (51.0%) females. The median serum UA level was 5.27 mg/dL in females and 6.84 mg/dL in males (Figure 2). The baseline characteristics of all patients, and of females and males separately, are listed in Table 1. Hyperuricemia was defined as a UA level > 5.6 mg/dL in females and >7 mg/dL in males. Of all hyperuricemic patients, 108 (42.6%) were male, and 95 (35.9%) were female. The baseline clinical characteristics of all patients are shown in Table 2. No significant difference in age, alcohol consumption, smoking status, or hypertension status was seen between hyperuricemic patients and those with normal UA levels. The prevalence of diabetes mellitus was significantly higher in hyperuricemic patients, including body mass index and total cholesterol, triglyceride, and creatinine levels. The high-density lipoprotein cholesterol level and eGFR were lower in both male and female hyperuricemic patients than in the patients with normal UA levels. The systolic/diastolic blood pressure was significantly elevated in females, but not males, with hyperuricemia compared to patients with normal UA levels. Female, but not male, hyperuricemic patients exhibited significantly lower serum albumin and higher serum IgA level than patients with normal UA levels. Table 1 summarizes the clinical characteristics of males and females with normal and elevated UA levels.

### 3.2. Histopathological Characteristics According to UA Levels in Males and Females

The histopathological findings of hyperuricemic patients and those with normal UA levels are shown in Table 3. Hyperuricemic females exhibited significantly higher rates of GS, SS, and CA (*p* < 0.001, *p* = 0.001, and *p* = 0.0.002, respectively). The MME (*p* = 0.023), IF (*p* < 0.001), and TA (*p* < 0.001) scores were also significantly higher in female hyperuricemic patients than in those with normal UA levels. Hyperuricemic males also exhibited significantly higher GS, SS, and CA scores than those with normal UA levels (*p* = 0.004, *p* = 0.002 and *p* = 0.004, respectively). The MME (*p* = 0.037), ECP (*p* = 0.001), IF (*p* < 0.001), and TA (*p* < 0.001) scores were also significantly higher in males with than without hyperuricemia. Immunofluorescence microscopy revealed that the mesangial IgA, C3, and C4d grades were not associated with hyperuricemia in either females or males. The WHO IgAN grades were significantly higher in both female and male hyperuricemic patients than those with normal UA levels (*p* < 0.001, *p* = 0.002, respectively).

### 3.3. Associations between Serum UA Levels and Histopathological Parameters

We used linear regression analysis to test for associations between the UA level and histopathological severity (Table 4 and Table 5). For all patients, the GS (*p* < 0.001), SS (*p* < 0.001), MME (*p* < 0.001), ECP (*p* < 0.001), IF (*p* < 0.001), TA (*p* < 0.001), and C3 mesangial deposition (*p* = 0.008) scores were positively correlated with the serum UA level in univariate analysis, as were the GS (*p* = 0.002), SS (*p* = 0.004), MME (*p* < 0.001), ECP (*p* = 0.026), IF (*p* < 0.001), TA (*p* < 0.001), and C3 mesangial deposition (*p* = 0.037) scores in multivariate analysis. In females, the GS (*p* < 0.001), SS (*p* < 0.001), CA (*p* = 0.003), MME (*p* = 0.001), ECP (*p* < 0.001), IF (*p* < 0.001), TA (*p* < 0.001), and C3 mesangial deposition (*p* = 0.036) scores positively correlated with the serum UA level in univariate analysis. In multivariate analysis, the MME (*p* = 0.002), ECP (*p* = 0.001), IF (*p* = 0.023), and TA (*p* = 0.002) scores were positively associated with the serum UA level in females. In contrast, the GS (*p* = 0.005), SS (*p* = 0.013), CA (*p* = 0.002), ECP (*p* = 0.014), IF (*p* < 0.001), and TA (*p* < 0.001) scores were positively associated with the serum UA level of males in univariate analysis. However, multivariate analysis revealed no significant association between any histopathological parameter and the serum UA level in males.

## 4. Discussion

This study showed that the UA level is associated with poor histopathological findings, such as MME, ECP, IF, and TA in female IgAN patients, but not males. Hyperuricemia is becoming more common, attributed to lifestyle and dietary changes and the aging of the population [27]. There is no consensus threshold serum UA level for hyperuricemia diagnosis. In addition, serum UA levels differ between males and females. We defined hyperuricemia as a serum UA level >7.0 mg/dL for males and >5.6 mg/dL for females [23].

To evaluate the prognosis of IgAN, WHO [26], Haas [28], and Oxford [29] classification have been used. The key histological damage is reflected in the MCP(M), ECP(E), GS(S), TF(T) and cresent(C) [30]. MEST-C scoring system was recently used for evaluation of IgAN prognosis. However, the MEST-C scoring system roughly reflects the extent of tissue damage in a way that scores with 0 and 1 (M, E, S) or 0, 1, and 2 (T) for each item. For this reason, we analyze key histological damage mentioned above with a more specialized hospital consensus scoring system [31].

A few studies have investigated the UA levels and histological findings of IgAN patients, but they focused only on the TA and IF scores [20,23,32]. Recent studies reported that mesangial C3 deposition was associated with active inflammation and poor outcomes for IgAN patients [33,34]. Hyperuricemia and C3 deposition were independent risk factors for IgAN, as were the Oxford T-score and a declining eGFR [35]. In the univariate analysis conducted in the present study, mesangial C3 deposition was correlated with the serum UA level of female patients. UA may activate the complement system and cause inflammation in association with IgAN.

In our study, the GS, SS, MME, ECP, IF, and TA scores were higher in both male and female hyperuricemic patients than in patients with normal UA levels. In the linear regression analysis, the GS, SS, MME, ECP, IF, TA and C3 mesangial deposition scores were associated with the UA level in all patients and in females only; however, only the GS, CA, IF, and TA scores showed an association with the UA level in males. The MME, ECP, IF, and TA scores were correlated with the serum UA level in females but not in males. Oh, et al. found that the serum UA level was an independent risk factor for IgAN progression, particularly in females [22]. However, histological findings were not analyzed.

Regarding how UA could affect the kidney, hyperuricemia can lead to GS, interstitial injury, and fibrosis [36,37], as well as oxidative stress. Many factors are involved in endothelial dysfunction, renal arteriopathy, and renovascular constriction [16,38,39,40]. It remains unclear why the serum UA level is more important in females than males, although it is known that estrogen inhibits urate transporter 1, reduces the serum UA level, and promotes urinary UA excretion [41]. The serum UA level increases in postmenopausal females, while the level of estrogen and of the transcriptional factors that it, directly and indirectly, regulates decreases [13]. Estrogen plays a crucial role in reno-protection. Estrogen negatively regulates TGF-ß synthesis; estrogen deficiency and ovariectomy accelerate the progression of glomerular injury and may contribute to the observed gender differences in IgAN [42]. Endogenous estradiol was shown to reduce the serum UA level [43].

This study had several limitations. First, it used a cross-sectional, retrospective design, so there were no long-term follow-up data. Second, the study involved eight university hospitals, and renal biopsy specimens were reviewed by eight different renal pathologists; thus, there may have been a degree of interobserver variability. However, all hospitals used identical biopsy criteria, and all pathologists were trained by the same international expert.

## 5. Conclusions

The serum UA level may affect more severe renal histopathology of female IgAN patients than male patients. A large prospective study is needed to validate this and to assess whether appropriate management of the UA levels of female patients can prevent IgAN-related histological damage.

## Figures and Tables

**Figure 1 jcm-10-01885-f001:**
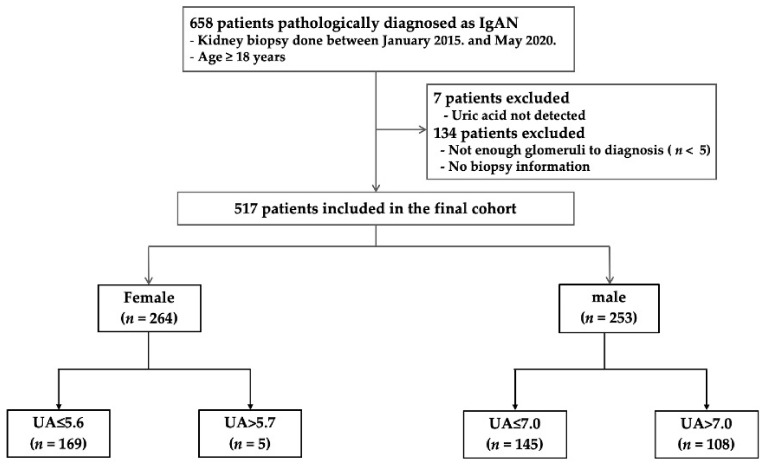
Flow diagram for the study population.

**Figure 2 jcm-10-01885-f002:**
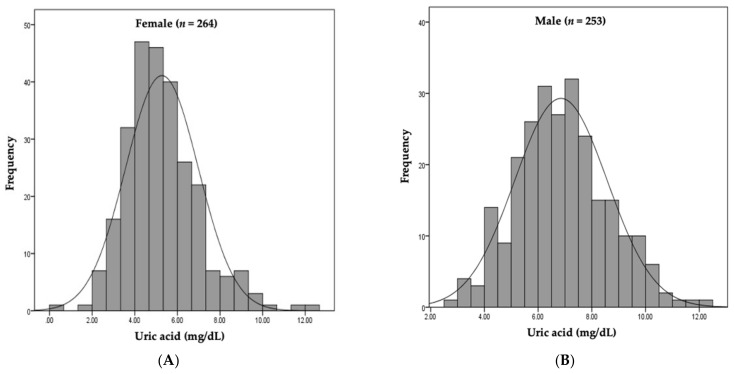
Distribution of serum uric acid by sex. **(A**) Median serum UA level was 5.27 mg/dL in females. (**B**) Median serum UA level was 6.84 mg/dL in males.

**Table 1 jcm-10-01885-t001:** Baseline clinical variables of the total patient and each sex group at the time of renal biopsy.

	Total (*n* = 517)	Female (*n* = 264)	Male (*n* = 253)
Uric acid (mg/dL)	6.04 ± 1.88	5.27 ± 1.70	6.84 ± 1.72
Age	42.02 ± 14.68	42.37 ± 12.95	41.66 ± 16.31
BMI (kg/m^2^)	24.10 ± 3.98	23.68 ± 4.19	24.53 ± 3.71
Alcohol (yes, %)	92 (17.8)	27 (10.3)	65 (25.7)
Smoking (yes, %)	71 (13.8)	9 (3.4)	62 (24.5)
DM (%)	31 (6.0)	16 (6.1)	15 (5.9)
HTN (%)	167 (33.3)	79 (30.9)	88 (35.9)
Systolic BP (mmHg)	124.76 ± 16.28	122.28 ± 16.94	127.36 ± 15.16
Diastolic BP (mmHg)	76.36 ± 10.12	75.26 ± 9.82	77.51 ± 10.33
Hemoglobin (g/dL)	13.14 ± 1.95	12.34 ± 1.61	13.98 ± 1.93
hs-CRP (mg/dL)	0.59 ± 3.46	0.42 ± 1.46	0.75 ± 4.71
Serum glucose (mg/dL)	107.5 ± 31.65	108.94 ± 35.43	106.13 ± 27.24
Serum creatinine (mg/dL)	1.14 ± 0.92	0.98 ± 0.83	1.30 ± 0.99
eGFR (mL/min/1.73 m^2^)	81.86 ± 33.95	83.39 ± 33.56	80.25 ± 34.35
Serum albumin (g/dL)	4.01 ± 0.58	3.98 ± 0.53	4.04 ± 0.63
AST (IU/L)	23.31 ± 11.4	21.88 ± 10.99	24.80 ± 11.65
ALT (IU/L)	22.03 ± 24.6	17.69 ± 13.13	26.54 ± 32.00
Total cholesterol (mg/dL)	186.25 ± 49.8	192.09 ± 53.42	180.20 ± 45.12
Triglyceride (mg/dL)	153.95 ± 123.71	135.04 ± 92.71	173.54 ± 146.84
LDL-C (mg/dL)	107.67 ± 39.29	106.82 ± 35.88	108.56 ± 42.64
HDL-C (mg/dL)	52.17 ± 16.71	53.16 ± 17.82	51.15 ± 15.45
Urine P/Cr (mg/mg)	1.53 ± 2.49	1.37 ± 2.01	1.70 ± 2.90
Urine RBCs (grade)	2.85 ± 1.54	2.81 ± 1.59	2.89 ± 1.49
Serum IgA (mg/dL)	312.08 ± 147.8	307.51 ± 134.8	316.77 ± 160.34

BMI, body mass index; SBP, systolic blood pressure; DBP, diastolic blood pressure; hs-CRP, high-sensitivity-C-reactive protein; eGFR, estimated glomerular filtration rate; AST, aspartate aminotransferase; ALT, alanine aminotransferase; LDL-C, low-density lipoprotein cholesterol; HDL-C, high-density lipoprotein cholesterol; RBCs, red blood cells; C3, complement 3; C4, complement 4.

**Table 2 jcm-10-01885-t002:** Baseline clinical variables of the uric acid group at the time of renal biopsy. Comparison between normal UA and hyperuricemia in each female and male group measured.

	Uric Acid					
	Female UA ≤ 5.6 *n* = 169	Female UA > 5.6 *n* = 95	*p* Value ^#^	Male UA ≤ 7.0 *n* = 145	Male UA > 7.0 *n* = 108	*p* Value ^##^
Uric acid (mg/dL)	4.28 ± 0.91	7.03 ± 1.33	<0.001	5.67 ± 0.98	8.43 ± 1.12	<0.001
Age	42.15 ± 12.30	42.76 ± 14.08	0.712	41.58 ± 17.27	41.75 ± 15.02	0.934
Sex (%)	169 (64)	95 (35.9)		145 (57.3)	108 (42.7)	
BMI (kg/m^2^)	23.09 ± 3.49	24.75 ± 5.07	0.002	23.97 ± 3.81	25.8 ± 3.45	0.005
Alcohol (yes, %)	20 (11.8)	5 (7.4)	0.261	34 (23.4)	31 (28.7)	0.344
Smoking (yes, %)	6 (3.6)	3 (3.2)	0.333	36 (24.8)	26 (24.1)	0.959
DM (%)	6 (3.6)	10 (10.6)	0.021	10 (6.9)	5 (4.6)	0.450
HTN (%)	54 (32.9)	25 (27.2)	0.339	51 (36.2)	37 (35.6)	0.924
Systolic BP (mmHg)	118.76 ± 16.17	128.62 ± 16.53	<0.001	126.90 ± 15.64	128.00 ± 14.54	0.571
Diastolic BP (mmHg)	73.80 ± 9.94	77.89 ± 9.07	0.001	77.25 ± 9.63	77.86 ± 11.25	0.643
Hemoglobin (g/dL)	12.49 ± 1.30	12.11 ± 2.04	0.088	14.07 ± 1.86	13.86 ± 2.03	0.399
hs-CRP (mg/dL)	0.33 ± 0.98	0.60 ± 2.06	0.151	0.88 ± 5.94	0.58 ± 2.01	0.623
Serum glucose (mg/dL)	109.80 ± 35.76	107.33 ± 34.94	0.597	105.78 ± 31.20	106.58 ± 21.00	0.819
Serum creatinine (mg/dL)	0.77 ± 0.48	1.36 ± 1.13	<0.001	1.12 ± 1.02	1.55 ± 0.89	0.001
eGFR (mL/min/1.73 m^2^)	95.74 ± 29.24	61.18 ± 29.20	<0.001	91.44 ± 32.75	64.95 ± 30.48	<0.001
Serum albumin (g/dL)	4.06 ± 0.49	3.85 ± 0.59	0.003	4.11 ± 0.63	3.95 ± 0.63	0.051
AST (IU/L)	21.69 ± 11.00	22.23 ± 11.02	0.705	24.55 ± 11.11	25.12 ± 12.38	0.700
ALT (IU/L)	17.53 ± 14.05	17.98 ± 11.37	0.789	26.70 ± 38.39	26.33 ± 20.81	0.928
Total cholesterol (mg/dL)	186.04 ± 45.34	203.14 ± 64.49	0.013	175.23 ± 40.33	186.79 ± 50.21	0.044
Triglyceride (mg/dL)	126.05 ± 81.65	151.46 ± 108.64	0.034	149.41 ± 115.07	205.49 ± 176.06	0.003
LDL-C (mg/dL)	107.06 ± 39.54	117.12 ± 47.9	0.075	101.08 ± 31.67	109.22 ± 38.73	0.071
HDL-C (mg/dL)	58.47 ± 19.41	53.51 ± 15.28	0.036	50.73 ± 13.82	43.20 ± 11.93	<0.001
Urine P/Cr (mg/mg)	1.12 ± 1.79	2.68 ± 4.14	<0.001	1.11 ± 1.46	1.73 ± 2.35	0.013
Urine RBCs (grade)	2.89 ± 1.53	2.76 ± 1.55	0.514	2.90 ± 1.57	2.78 ± 1.51	0.558
Serum IgA (mg/dL)	285.74 ± 97.25	333.46 ± 165.03	0.004	308.97 ± 129.20	338.48 ± 204.68	0.169

BMI, body mass index; SBP, systolic blood pressure; DBP, diastolic blood pressure; hs-CRP, high-sensitivity-C-reactive protein; eGFR, estimated glomerular filtration rate; AST, aspartate aminotransferase; ALT, alanine aminotransferase; LDL-C, low-density lipoprotein cholesterol; HDL-C, high-density lipoprotein cholesterol; RBCs, red blood cells. ^#^ female UA ≤ 5.6 vs. female UA > 5.6. ^##^ male UA ≤ 7.0 vs. male UA>7.0.

**Table 3 jcm-10-01885-t003:** Histopathological characteristics between the uric acid group.

	Uric Acid						
		Female UA ≤ 5.6	Female UA > 5.6	*p* Value ^#^	Male UA ≤ 7.0	Male UA > 7.0	*p* Value ^##^
	**Light microscopy**						
Global sclerosis (%)		13.96 ± 15.12	25.86 ± 24.84	<0.001	13.85 ± 16.44	21.09 ± 22.70	0.004
Segmental sclerosis (%)		7.31 ± 11.99	13.66 ± 16.99	0.001	6.94 ± 9.67	11.68 ± 13.84	0.002
Capsular adhesion (%)		7.31 ± 11.99	13.66 ± 16.99	0.002	6.84 ± 10.32	11.49 ± 14.63	0.004
Mesangial matrix expansion (0–4)		1.97 ± 0.89	2.23 ± 0.83	0.023	2.10 ± 0.87	2.32 ± 0.78	0.037
Mesangial cell proliferation (0–4)		2.01 ± 0.88	2.17 ± 0.86	0.162	2.15 ± 0.83	2.23 ± 0.84	0.444
Endocapillary proliferation (0–4)		0.11 ± 0.40	0.25 ± 0.76	0.058	0.08 ± 0.33	0.29 ± 0.64	0.001
Interstitial fibrosis (0–4)		1.08 ± 0.86	1.67 ± 1.03	<0.001	1.22 ± 0.95	1.77 ± 0.93	<0.001
Tubular atrophy (0–4)		1.04 ± 0.86	1.62 ± 1.07	<0.001	1.22 ± 0.95	1.72 ± 0.94	<0.001
	**Immunofluorescence microscopy**						
IgA Mesangial deposit (0–4)		3.24 ± 0.95	3.27 ± 1.11	0.794	3.36 ± 0.84	3.35 ± 0.98	0.895
C3 Mesangial deposit (0–4)		1.97 ± 1.18	2.18 ± 1.35	0.199	2.20 ± 1.10	2.26 ± 1.05	0.676
C4d Mesangial deposit (0–4)		0.02 ± 0.22	0.02 ± 0.20	0.882	0.05 ± 0.32	0.05 ± 0.33	0.940
	**WHO classification (*n* = 437)**						
Grade (1–6)		2.83 ± 0.72	3.38 ± 0.83	<0.001	2.86 ± 0.83	3.25 ± 0.97	0.002

C3, complement 3; C4, complement 4; IgA, immunoglobulin. ^#^ female UA ≤ 5.6 vs. female UA > 5.6. ^##^ male UA ≤ 7.0 vs. male UA>7.0.

**Table 4 jcm-10-01885-t004:** Linear regression for uric acid and the histopathologic parameters in all patients.

			Univariable				Multivariable	
	β	t	r^2^	*p* Value	β	t	r^2^	*p* Value
Global sclerosis	0.258	6.071	0.001	<0.001	0.146	3.075	0.059	0.002
Segmental sclerosis	0.166	3.812	0.001	<0.001	0.130	2.860	0.087	0.004
Mesangial matrix expansion	0.172	3.950	0.007	<0.001	0.168	3.6660	0.031	<0.001
Capsular adhesion	−0.003	−0.071	0.000	0.943	-	-	-	-
Endocapillary proliferation	0.200	4.623	0.001	<0.001	0.164	2.240	0.037	0.026
Interstitial fibrosis	0.332	7.951	0.001	<0.001	0.218	4.743	0.023	<0.001
Tubular atrophy	0.340	8.169	0.001	<0.001	0.260	5.452	0.030	<0.001
IgA mesangial deposit	0.003	0.078	0.000	0.938	-	-	-	-
C3 mesangial deposit	0.118	2.680	0.002	0.008	0.093	20.90	0.027	0.037
C4 mesangial deposit	0.074	1.679	0.002	0.094	-	-		-

Multivariable analysis was adjusted for clinical parameters, including age, DM, systolic BP, BMI, Hb, total cholesterol, eGFR, urine P/C and IgA level.

**Table 5 jcm-10-01885-t005:** Linear regression for uric acid and the histopathologic parameters in sex differences (a) univariable analysis (b) multivariable analysis.

(a)								
	Univariable							
	Female				Male			
	β	t	r^2^	*p* Value	β	t	r^2^	*p* Value
Global sclerosis	0.359	6.184	0.129	<0.001	0.176	2.811	0.031	0.005
Segmental sclerosis	0.225	3.730	0.025	<0.001	0.157	2.500	0.025	0.013
Mesangial matrix expansion	0.208	3.444	0.043	0.001	0.097	1.542	0.009	0.124
Capsular adhesion	0.187	2.996	0.035	0.003	0.146	2.297	0.017	0.022
Endocapillary proliferation	0.268	4.488	0.072	<0.001	0.155	2.485	0.024	0.014
Interstitial fibrosis	0.392	6.886	0.154	<0.001	0.263	4.280	0.069	<0.001
Tubular atrophy	0.396	6.970	0.157	<0.001	0.265	4.317	0.070	<0.001
IgA mesangial deposit	0.030	0.483	0.001	0.629	−0.080	−1.269	0.006	0.206
C3 mesangial deposit	0.130	2.111	0.017	0.036	0.052	0.822	0.003	0.412
C4 mesangial deposit	0.055	0.885	0.003	0.377	−0.026	−0.398	0.001	0.691
**(b)**								
	**Multivariable**							
	**Female**				**Male**			
	**β**	**t**	**r^2^**	***p* Value**	**β**	**t**	**r^2^**	***p* Value**
Global sclerosis	0.112	1.689	0.316	0.092	-	-		-
Segmental sclerosis	0.584	0.044	0.171	0.573	-	-		-
Mesangial matrix expansion	0.189	2.989	0.036	0.003	-	-		-
Capsular adhesion	-	-	-	-	-	-		-
Endocapillary proliferation	0.224	3.367	0.117	0.001	-	-		-
Interstitial fibrosis	0.155	2.282	0.301	0.023	0.093	1.432	0.357	0.154
Tubular atrophy	0.188	2.769	0.304	0.006	0.102	1.581	0.354	0.115
IgA mesangial deposit	-	-	-		−0.105	−1.623	0.063	0.106
C3 mesangial deposit	-	-	-			-	-	-
C4 mesangial deposit	-	-	-			-	-	-

Multivariable analysis was adjusted for clinical parameters, including age, DM, systolic BP, BMI, Hb, total cholesterol, eGFR, urine P/C and IgA level.

## Data Availability

All data are reported in the article.

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
