# Peer review of "The Serum Uric Acid Level Is Related to the More Severe Renal Histopathology of Female IgA Nephropathy Patients"

_jcm, 2021, doi:10.3390/jcm10091885_

Round 1

Reviewer 1 Report

Title: The serum uric acid level is related to the prognosis of female IgA nephropathy patients. There are my suggestions, which should be taken into account, before further processing of the manuscript.

IgA nephropathy is the most frequent primary glomerulonephritis worldwide and its presentation varies from rather mild symptoms to nephrotic range proteinuria and, in rare cases, to rapid progressive glomerulonephritis. Due to such heterogeneity it is of utmost importance to find both clinical and histopathological predictors of disease outcome. Therefore I found a paper about potential predictors of IgAN progression of highest importance. 

1. My remark is the use of WHO classification instead of Oxford classification of IgAN but results based on both classifications are comparable.

2. The title is misleading. Authors wrote about IgA nephropathy outcome but based on the presented results, there is nothing about outcome. Authors did not assume any endpoints and as a consequence it is hard to say anything about prognosis.

3. The other problem considers the novelty of the paper. Through my research in PubMed i found several publications considering the role of serum uric acid concentration and their influence on IgA nephropathy outcome also taking into account sex differences. Therefore I would like to ask Authors to explain the novelty of their paper.

Reviewer 2 Report

This paper describes the association between uric acid levels at time of kidney biopsy with clinicopathological features at time of diagnosis.

I have a number of comments:

  1. The title of the paper mentions prognosis- there is no assessment of prognosis in this paper- it is simply describing associations at time of kidney biopsy and does not link these to clinical outcome.
  2. Multiple statistical comparisons are performed on the same data set but no correction for multiple comparisons has been made- this needs to be corrected
  3. The scoring systems for kidney biopsy evaluation are outdated, not validated and not consistent with current standards- for this analysis the Oxford MEST-C should be used- it is internationally validated and the standard for reporting IgAN kidney biopsies.

Round 2

Reviewer 1 Report

I appreciate the changes you have made according to both Reviewers' suggestions. I especially acknowledge the correction of the paper's title, which is no longer misleading. Please change the typo in the title.